# Empirical Analysis of Autonomous Vehicle’s LiDAR Detection Performance Degradation for Actual Road Driving in Rain and Fog

**DOI:** 10.3390/s23062972

**Published:** 2023-03-09

**Authors:** Jiyoon Kim, Bum-jin Park, Jisoo Kim

**Affiliations:** Department of Highway & Transportation Research, Korea Institute of Civil Engineering and Building Technology, Goyang-si 10223, Gyeonggi-do, Republic of Korea; jiyoonkim@kict.re.kr (J.K.);

**Keywords:** LiDAR, weather condition, detection performance, detection indicator, empirical test

## Abstract

Light detection and ranging (LiDAR) is widely used in autonomous vehicles to obtain precise 3D information about surrounding road environments. However, under bad weather conditions, such as rain, snow, and fog, LiDAR-detection performance is reduced. This effect has hardly been verified in actual road environments. In this study, tests were conducted with different precipitation levels (10, 20, 30, and 40 mm/h) and fog visibilities (50, 100, and 150 m) on actual roads. Square test objects (60 × 60 cm^2^) made of retroreflective film, aluminum, steel, black sheet, and plastic, commonly used in Korean road traffic signs, were investigated. Number of point clouds (NPC) and intensity (reflection value of points) were selected as LiDAR performance indicators. These indicators decreased with deteriorating weather in order of light rain (10–20 mm/h), weak fog (<150 m), intense rain (30–40 mm/h), and thick fog (≤50 m). Retroreflective film preserved at least 74% of the NPC under clear conditions with intense rain (30–40 mm/h) and thick fog (<50 m). Aluminum and steel showed non-observation for distances of 20–30 m under these conditions. ANOVA and post hoc tests suggested that these performance reductions were statistically significant. Such empirical tests should clarify the LiDAR performance degradation.

## 1. Introduction

### 1.1. Challenges of Autonomous Vehicles

Autonomous vehicles (AVs) use sensors to collect road environment information that is then used for vehicle control. AV sensors play the same role as the driver’s eyes in recognizing the road environment while driving. Various AV sensors, such as vision, light detection and ranging (LiDAR), radar, and ultrasonic sensors, are used, and each of these plays different roles depending on the detection area and target [1]. Although AV sensor performance has been significantly improved using data handling and artificial intelligence (AI) algorithms, it is insufficient for realizing the commercialization of AVs [2,3]. One factor hindering the commercialization of AVs is that the autonomous driving function does not operate accurately in some scenarios. Examples of such scenarios include bad weather, such as rain, fog, and snow. In a focus group interview conducted with vehicle experts and engineers, bad weather was identified as one of the main obstacles to overcome [4]. An analysis of an AV disengagement report in California, USA, suggested that a deterioration in the weather was the most frequent external causative factor [5]. This is because bad weather hampers an AV’s ability to judge a complicated road and degrades the sensor performance. However, a survey revealed that drivers desire autonomous driving technology most in situations in which they themselves cannot easily drive the vehicle, such as under severe weather conditions [6]. Therefore, overcoming bad weather is an important goal for realizing the commercialization of AVs [7].

Accordingly, recent studies have been investigating cooperative autonomous driving. Korea is planning and advancing cooperative autonomous driving through cooperation with infrastructure [8]. However, the construction of cooperative infrastructure such as communication facilities, traffic control centers, and road traffic facilities requires considerable time and is infeasible for all road sections. Therefore, AVs are expected to continue relying on sensor-derived information; in fact, AVs being developed currently still prioritize sensor information [1]. In addition, considering the occurrence of emergencies such as communication disconnection even after the cooperative infrastructure is established, sensors will remain important for the safe operation of AVs by ensuring the redundant collection of road-environment information [9].

### 1.2. LiDAR Sensors in AVs and Limitations

LiDAR-based sensors are the primary sensors used in AV technology [10]. LiDAR involves the irradiation of a laser, with near-infrared wavelengths, thousands of times per second onto an object. It then detects the reflected light and, based on the time measurement, determines the distance to the object with high resolution and accuracy. LiDAR is widely used in AV technology for several reasons. First, it can be used during nighttime without restrictions because it is less affected by illuminance changes. Second, it is more robust than a vision sensor under bad weather conditions, including rain, snow, and fog [11]. Third, it can identify the geometry of a detected object in 3D and can measure the distance to it with high accuracy. Based on these advantages, LiDAR is used for detecting static [10] and dynamic objects, such as pedestrians and vehicles, on the road [12]. LiDAR is also used to detect traffic safety facilities, such as lanes and road signs, and to correct the localization of AVs through map matching with a precise HD map for AVs [13]. However, LiDAR sensors are costly, and a mechanical-type LiDAR that rotates the transceiver fails frequently. Despite these disadvantages, LiDAR has contributed to the rapid development of AVs and has been used for obtaining road information for safe driving [10]. For example, approximately 85% of AVs that gained an autonomous driving test certification from the Korean government were equipped with LiDAR [9]. In recent years, to overcome these disadvantages, a solid-type LiDAR that does not rotate has increasingly been used as an alternative [14]. Further, LiDAR is increasingly being combined with vision sensors [15]. In particular, the combination of vision and LiDAR exhibits better performance than that of vision alone [16]. Therefore, studies are actively investigating combinations of sensors to improve the driving safety of AVs [17].

Although LiDAR is more robust to climate change than vision sensors, an additional disadvantage is that it is affected by bad weather. Studies have demonstrated that LiDAR performance degrades in bad weather [12,13,18,19,20,21,22]. This performance degradation has been attributed to interference by moisture in the air [20,23]. The LiDAR performance degradation under bad weather has been demonstrated through simulation [23] or in a limited test site, such as a climate chamber [20]. However, few studies have verified the effect of rain and fog on LiDAR-detection performance at the same time and place on an actual road. To achieve the commercialization of AVs, both sensor development and continuous performance verifications are required.

### 1.3. Research Approach

Owing to the lack of studies on LiDAR sensor degradation in bad weather, the present study tested the degradation of LiDAR-detection performance during bad weather in an actual road environment. Quantifiable performance indicators were selected, and their variation (particularly degradation) in rain and fog were experimentally determined. Representative materials that are typically seen on Korean roads were used to manufacture sign panels of the same size as that of road signs and in compliance with Korea’s road regulations. Data of the performance indicators were obtained from a LiDAR sensor installed in an autonomous driving test vehicle certified by the government of South Korea. The test site was the Center for Road Weather Proving Ground, Korea Institute of Construction and Building Technology, where realistic rain and fog conditions are reproducible in actual road environments. The rain and fog conditions were controlled by adjusting the volume of rain (precipitation) and visibility distance, respectively. These empirical experiments conducted in rain and fog with various materials are expected to provide an understanding of the degradation of LiDAR-detection performance and to develop improved road facilities (road signs, etc.) for easy detection with LiDAR.

The structure of this paper is shown in Figure 1. For convenience of reading, the definition of key terms is presented in the Appendix A.

## 2. Literature Review

With the widespread commercialization of global navigation satellite system (GNSS) technology since the early 1990s, LiDAR has become a well-established surveying technique for acquiring geospatial information [24]. The miniaturized mobile LiDAR that is mainly used in AVs is the same as airborne/terrestrial LiDAR; however, it has been rapidly developed within a short period of time. LiDAR irradiates a laser on target objects and detects light reflected from them to measure the distances to these objects and, thereby, precisely describes the surroundings in 3D. LiDAR mainly uses a laser with wavelengths of 760–1900 nm in the near-infrared band. LiDAR is divided into the phase-shift and time-of-flight (TOF) methods depending on the irradiation method. Recently, the TOF method has been widely used [25].

Studies on the general characteristics of LiDAR are reviewed as follows. Because the laser pulses are emitted in layers with a set angle range, the detection accuracy of LiDAR decreases as the distance to the object increases [22]. Further, LiDAR exhibits higher performance when the object’s reflective surface is smoother. Lee [26] experimentally demonstrated that LiDAR exhibits high performance when the object is colored rather than black; in particular, it achieves the highest performance when the object is white. Kim and Park [21] confirmed that LiDAR exhibits the highest performance when detecting retroreflective films and white signs. Kim et al. [22] observed that retroreflective films, used as sign materials, showed the highest detection performance, followed by aluminum, steel, plastic, and wood. Kim and Park [9] divided the factors that affect LiDAR-detection performance into two categories. One category is the mechanical specifications, including the number of channels used, wavelength and intensity of the laser, vertical field of view (FOV), and points per second. These specifications are provided by each manufacturer and are, unfortunately, difficult to identify otherwise. The other category is factors related to the object to be detected, including the distance to the object as well as the material, color, shape, and arrangement of the object. LiDAR uses the near-infrared band; therefore, detecting objects without references to ambient light is possible [12].

LiDAR has strong positive characteristics, such as ‘sensor suitable for recognizing objects’, ‘excellent accuracy in detecting objects on the road’, and ‘maintaining its performance even at night owing to being less affected by light’ [12]. The latest AVs use LiDAR to collect information on dynamic objects, such as pedestrians, traffic lights, nearby vehicles, bicycles, and motorcycles [12]. For example, Google’s Waymo uses LiDAR for detecting unexpected objects [27]. To accurately classify and recognize objects from data detected by LiDAR, various models have been developed, and algorithm-development studies using semantic segmentation have been conducted [28,29]. Moreover, studies have been investigating the detectability of objects by LiDAR. For example, the Korea Institute of Civil Engineering and Building Technology [1] produced LiDAR-detectable signs by mixing black rubber, wood, and retroreflective films and tested them in considering that the LiDAR-detection performance varies depending on the object color. Further, Kim and Kim [30] produced a flat traffic cone instead of a rounded one to maximize the laser reflection and observed that the former exhibited improved LiDAR-detection performance compared with that of the latter. LiDAR can also be used for localization or positioning; this is essential for autonomous driving based on precise maps and for detecting obstacles or facilities. A method for the classification and localization of road signs in a 3D space using a neural network and point cloud obtained from LiDAR has been introduced [31]. A road-sign perception system for vehicle localization within a third-party HD map using LiDAR has also been proposed [32]. Localization is also used to recognize road facilities, such as nearby lanes and signs, and to correct GPS-based absolute localization through map matching with a pre-made intensity map [13].

Despite the high cost of LiDAR, its role and utilization in AVs are increasing because it enables high-distance accuracy and can collect information of a 3D space [10,33]. Further, LiDAR is less affected by illuminance and exhibits relatively robust characteristics even in rain and fog; in contrast, vision sensors are significantly affected by weather conditions including low illuminance, rain, fog, snow, and direct sunlight [11]. However, many studies have reported performance degradations in LiDAR with changes in the weather. Kutila et al. [19] confirmed through actual measurements that the detection performance (detection distance) of LiDAR is reduced by 25% in fog and snowfall conditions. However, the limitation of their study is that the quantitative analysis of the effect of changes in fog and snow on the deterioration of the detection performance of LiDAR was not possible because the information on the amount of snowfall or fog visibility was not accurate at the time of measurement. Goodin et al. [23] suggested through simulation that the maximum object recognition distance and number of point clouds (NPC) decreased with the increase in rainfall. At the maximum rainfall of 45 mm/h, the maximum recognition distance decreased by approximately 30% (5 m) compared with a clear day, and the NPC decreased by 45%. In addition, two reasons were summarized for this result as follows. The number of laser points (number of NPCs) is reduced because the laser reflected from the object collides with the raindrops, and the distance to the object is less accurate because the laser is reflected by the raindrops rather than the object. It is a significantly meaningful study in that it quantitatively suggests that the maximum LiDAR range, NPC data, and obstacle-detection range decrease with the increase in the amount of rainfall. Tang et al. [12] confirmed via a pedestrian-detection test conducted in a parking lot that the rainfall environment affects the detection accuracy of LiDAR. They estimated the probability of pedestrian detection failure in case of rain using a logit model. However, as in the case of Kutila et al. [19], the amount of rainfall was not measured, and the relationship between the actual amount of rainfall and detection performance of LiDAR could not be proven by classifying the meteorological factor only as ‘clear–rain’. Kim et al. [22] confirmed the detection performance of LiDAR with NPC and intensity. By measuring LiDAR data according to the distance to the sensing object, the material of the sensing object, rainfall, and speed of the car, the distance and rainfall affect NPC and intensity; the material affects intensity; and the speed does not affect the detection performance of LiDAR. Especially, they revealed that LiDAR cannot easily detect signs when the precipitation is 40 mm/h or higher, regardless of the material. This study is significant in that it expanded and analyzed four factors, whereas most studies set one factor influencing the detection. However, it has a limitation that the correlation between all factors could not be confirmed. Other studies observed that the detection performance also decreased under fog and snow conditions for reasons similar to those in the case of rain [18,34]. Kim et al. [22] observed that the LiDAR performance differs depending on the material even under the same precipitation. The differences between and limitations of the existing studies that analyzed the degradation of detection performance of LiDAR due to changes in weather conditions are summarized in Table 1.

LiDAR has mainly been used for detecting obstacles and nearby vehicles. In the future, it is expected to be used for providing dedicated road transportation facilities and localization information. Previous studies revealed that the LiDAR-detection performance is affected by the material or color of the target object, and it can be significantly affected by weather conditions; all of these are external influence factors. In particular, LiDAR performance was found to be significantly reduced by rain and fog, which frequently occur in reality. Therefore, the LiDAR-detection performance in AVs and factors influencing the same need to be investigated actively [18]. In particular, no study has investigated the influence of rain and fog on a LiDAR system installed in an actual AV in a same road environment. Under rain and fog, the LiDAR-detection performance needs to be accurately investigated in terms of changes in the values of objective performance indicators.

## 3. Methods

### 3.1. Test Site

Empirical experiments were conducted at Road Weather Proving Ground (PG). PG is equipped with weather environment reproduction facilities. It contains a straight multilane asphalt-paved road with a length of more than 1 km, as shown in Figure 2. In the right-hand side lane, a rainfall condition of up to 50 mm/h can be reproduced. In addition, the inside of the structure is equipped with a visibility sensor and a fog generation system. The system controls the fog intensity using glycerin and the visibility sensor. Therefore, the performance can be verified under different weather conditions in a realistic road environment. Point cloud data (PCD) were recorded by driving while varying rain and fog conditions.

### 3.2. Test Equipment and Objects

A Hyundai Tucson was used as the test vehicle; this vehicle had been modified to be equipped with autonomous driving functions by the Korea Institute of Civil Engineering and Building Technology, and it achieved a Korean autonomous driving test certification in 2022. This AV was equipped with LiDAR, radar, and vision sensors, as shown in Figure 3. This study aims to verify the LiDAR-detection performance; for this purpose, the data acquired using the 32 ch LiDAR installed on the vehicle roof (height: approximately 1.8 m) were analyzed.

Table 2 shows the specifications of a representative LiDAR sensor used for an Autonomous Vehicle. Among the sensors, Robosense’s RS-32 was used as the LiDAR sensor in the experiment. It is a rotation type LiDAR having 32 Channels of laser and uses 905 nm wavelength. As listed in Table 2, rotation-type LiDAR sensors have similar characteristics (wavelength, angular resolutions, points per second, etc.). Therefore, the experimental results may represent those of other rotation-type sensors although only Robosense’s RS-32 sensor was tested in this study. A solid-state-type sensor (Velarray in Table 2) uses almost the same wavelength as those used by other rotation-type sensors; therefore, a similar performance degradation phenomenon is expected to occur in a solid-state-type sensor.

In this experiment, the LiDAR sensor’s rotation speed was set to 10 Hz. Horizontal angular resolution was set to 0.2°. Vertical FOV was −25° to +15°, and it can generate 600,000 points in a second.

Square-shaped target objects with dimensions of 60 cm × 60 cm were manufactured for the experiment as this shape is used frequently for road signs in Korea. The objects were attached to a frame erected on the right-hand side of the vehicle driving path, as shown in Figure 4. The materials of the tested target objects were white retroreflective film, aluminum, steel, plastic, and black sheet.

### 3.3. Test Scenario

Table 3 lists the various factors and control conditions. Target objects made of five different materials—white retroreflective film, aluminum, steel, black sheet, and plastic—were tested. With regard to weather conditions, clear, four rainfall rates—10, 20, 30, and 40 mm/h—and three fog intensities (i.e., visible range)—50, 100, and 150 m—were tested. The distance between the object and LiDAR (detection distance) was varied as 10, 20, 30, 40, and 50 m. For each combination of detection distance and weather condition, the PCD of the object were collected through five drives for each experimental scenario.

### 3.4. Test Performance Indicators and Performance Verification Method

The NPC and intensity are frequently used indicators in LiDAR-related research [22,23]. LiDAR creates 3D point information of the surrounding environment by using an infrared laser pulse. When a laser pulse is irradiated from the transmitter toward the object and returned to the receiver after being reflected, a single point is created. These point data are accumulated to form a point cloud. In this study, NPC refers to the number of points in the cloud created from the target object by LiDAR scanning. NPC reflects whether the LiDAR is detecting objects adequately because when the points are acquired over a certain scale, LiDAR can be used for 3D object identification and classification through clustering. Therefore, as the NPC increases, the clustering of point clouds becomes easier, and the geometry of objects can be identified more accurately.

The intensity refers to the return strength of the laser pulse emitted from the LiDAR [31]. It is generally expressed using a number between 0 and 1 or between 0 and 255 (8-bit range), where 0 implies that no emitted laser pulse was returned at the laser receiver, and 1 or 255 implies that a very strong laser pulse was returned at the receiver. The intensity may vary depending on various factors, such as the pulse width of the irradiated laser; incidence angle on the object; distance from the object; surface material, color, and roughness of the object; and humidity [1,20,21,22,31,36].

The selected LiDAR performance indicators, namely, NPC and intensity, both depended on the detection distance (distance between the target object and LiDAR). For example, if the target object is a black rod, as shown in Figure 5, the NPC is higher when the distance between the object and LiDAR is 10 m than when it is 20 m. This difference increases with increasing distance between the object and LiDAR. Therefore, the performance for each variable was compared for the same detection distance for the test scenario described in Table 3.

The LiDAR performance was verified in the following order. First, point cloud plots for each object were drawn to identify the detected geometry according to the weather conditions and to explore the changes in performance indicators. The point cloud plot describes the data acquired on a 2D plane; therefore, it can show all geometries of each test scenario that are not represented only by numeric values of indicators. Subsequently, quantitative values of the NPC and intensity for each object were calculated and compared. Through this, the pattern of indicator value changes for each material and weather condition was explored, and the degree of decrease during bad weather was revealed. Finally, through ANOVA, whether the degradation in performance indicators between weather conditions is statistically significant was tested. Because ANOVA only shows whether there are groups having different mean values compared to other groups, a post hoc test was performed to identify the weather conditions in which the performance indicators decreased compared to those under clear conditions.

## 4. Point Cloud Plots

### 4.1. Overview

The point cloud plot of each object (sign) in the PCD collected through the test was described to examine the overall influence of the characteristics of the material, detection distances, and weather conditions. The point cloud plots were expressed on a 2D plane by extracting the PCD of each sign. The intensity of each point was expressed using a color scale (low as blue, high as red), so that both the geometry and reflectivity of the points could be identified through the graph. Graphs were presented in the order of the white retroreflective film, aluminum, steel, black sheet, and plastic. This order is based on the reflectivity of each material; the higher the reflectivity, the lesser the effect of the detection distance and bad weather [20,21,22,31,36].

### 4.2. Point Cloud Plots According to Precipitation and Visibility Distance

#### 4.2.1. Retroreflective Film (RF)

Figure 6 shows the point cloud plot of the retroreflective film sign (RF) by detection distance (Y-axis) according to the precipitation (X-axis on the left-hand side) and visibility distance (X-axis on the right-hand side). The geometry of the sign better describes the actual sign as the detection distance decreases from 50 m to 10 m. The differences between weather conditions are not clearly shown although the precipitation and visibility distance change.

Unlike other objects with diffuse reflection surfaces, RFs have significantly high reflectivity, and almost every point was recorded to have an intensity of over 200 in this test. Accordingly, within a detection distance of 50 m, NPCs close to the maximum value of the LiDAR spec were found to be detected regardless of the weather.

#### 4.2.2. Aluminum (AL)

Figure 7 shows the aluminum sign (AL) point cloud plot according to the precipitation and visibility distance. As with RF shown above, the geometry of the actual object is described better as the detection distance decreases from 50 m to 10 m. However, unlike RF, AL tends to show fewer NPCs and more poorly describes the actual geometry of the sign as the precipitation increases and visibility distance decreases.

This tendency mainly appears with severely bad weather and a detection distance over 20 m. In the strongest rain (40 mm/h) and fog (visibility distance less than 50 m), the intensity and NPC reduce significantly, eventually resulting in non-observation. Nevertheless, because AL is a highly reflective material, its geometry is well described under conditions of a detection distance of 10 m, rainfall less than 30 mm/h, and visibility distance over 50 m.

#### 4.2.3. Steel (ST)

Figure 8 shows the point cloud plot of the steel sign (ST) according to the precipitation and visibility distance. The geometry of the sign, as observed through the point cloud plot, better describes the geometry of the actual sign as the detection distance decreases from 50 m to 10 m, as in the case of the other materials mentioned above. However, ST has a relatively lower reflectivity than that of RF or AL and is more affected by bad weather.

Unlike AL, which was robust to rain less than 30 mm/h and fog visibility over 50 m, ST results in non-observation even under better weather conditions.

In particular, when the detection distance is 30 m or more, NPC clearly is reduced as the precipitation increases. In addition, at a visibility distance of 50 m, no points could be observed with a detection distance of 20 m or more.

#### 4.2.4. Plastic (PL)

Figure 9 shows the point cloud plot of the plastic sign according to the precipitation and visibility distance. The geometry of PL through the point cloud plot does not adequately describe the actual sign, regardless of the detection distance. This is mainly because PL is transparent, and, therefore, the laser pulse irradiated from the LiDAR penetrates the object and is rarely reflected, unlike in the case of other materials.

Surprisingly, under rainy conditions, a more detailed geometry was described, in contrast to the poor recognizability under clear conditions. In addition, at a detection distance of 10 m, the NPC increased significantly compared with that under clear conditions. This seems attributable to the temporary increase in the reflectance of the surface of the sign caused by water droplets. Although the NPC temporarily increased under both rain and fog, it decreased as the precipitation increased and visibility decreased, as in the case of the other materials. Therefore, point clouds were hardly observed at detection distances above 30 m, even under good weather conditions.

#### 4.2.5. Black Sheet (BS)

Figure 10 shows the point cloud plot of the black sheet sign (BS) according to the precipitation and visibility distance. Unlike RF or metallic signs (i.e., AL and ST), the geometry of the sign as observed through the point cloud plot is not adequately described, regardless of the detection distance. This is because the black color and sheet surface are both less reflective, and, therefore, the reflectance of the LiDAR laser pulse is reduced, as reported previously [20,21,22,31,36].

Expectedly, BS was strongly affected by bad weather. Even for the shortest detection distance (10 m), no point was observed for precipitation of 20 mm/h or more. Under fog, even with the best visibility condition (i.e., visibility distance of less than 150 m), LiDAR could detect the BS at all. However, with precipitation of 10 mm/h, NPC significantly increased compared with that under clear conditions for a detection distance of up to 30 m. This is attributed to the temporary increase in the reflectivity of the surface of the sign caused by water droplets.

## 5. Changes in Performance Indicators

### 5.1. Overview

In this section, the changes in the NPC and intensity depending on the material and weather conditions are presented. Figure 11 shows the NPC and intensity values measured at the same detection distance for easy comparisons with other materials and weather conditions. Table 4 and Table 5 list descriptive statistics and more detailed test results for the NPC and intensity, respectively.

The NPC and intensity are correlated to each other. As the reflectivity of the object surface decreases under bad weather conditions, the intensity first decreases, and once it decreases sufficiently, the NPC decreases. As such, the NPC and intensity are shown in parallel in Figure 11 to simultaneously examine the changes depending on the weather conditions and material. Table 4 and Table 5 list the values for all weather conditions, and through the ratio field, the decrease in NPC and intensity with the weather conditions can be compared with the values under clear conditions.

### 5.2. Performance Indicators According to Precipitation and Visibility Distance

As shown in Figure 11, RF exhibited the best results for both NPC and intensity, followed by AL, ST, PL, and BS. This is basically attributable to the reflectivity of each material. In addition, NPC decreased with an increase in the detection distance or worsening of the weather (i.e., increased precipitation or decreased visibility distance).

RF showed the highest NPC at all detection distances and under most weather conditions. This is attributable to its exceptionally high reflectivity; the intensity graphs indicate that most of the laser pulse emitted from LiDAR is reflected. RF could be seen under all weather conditions and detection distances, unlike other materials for which the NPC decreased appreciably or no point clouds were observed as the weather deteriorated.

However, even for RF, the values of indicators decreased owing to weather conditions. As seen in the ratio field in Table 4, the weather condition that reduced the detection performance the most was fog with a visibility distance less than 50 m. At a detection distance of 50 m with the thickest fog, the NPC decreased by 26%. At most other detection distances and weather conditions, the performance indicators remained at approximately 90% of those under clear conditions, implying that the performance degradation is significantly reduced by the high reflectivity of RF.

AL showed no significant difference in NPC between weather conditions at short detection distances (20 m or less), indicating its high reflectivity. However, as the detection distance increased to 50 m, the NPC noticeably decreased (in Figure 11), and non-observation occurred under bad weather conditions (gray shades in Table 4). Table 4 shows the decreased rate of NPC owing to weather conditions. At a short detection distance (10 m), the decrease was nonsignificant. However, at a detection distance of 20 m, NPC decreased by 81–86% when fog occurred. From a detection distance of 30 m, non-observation of the point cloud occurred (N < 5, gray shades) in intense fog (visibility distance less than 50 m) and rain (precipitation of 40 mm/h). Table 5 clearly shows that the intensity of AL decreases with weather conditions. Overall, the detection performance (mainly NPC) is maintained relatively adequately for AL signs, although the intensity value decreased under severe weather conditions.

ST showed a lower NPC overall than that of AL. As the weather deteriorated, NPC decreased non-significantly for a detection distance of 10 m and noticeably for a detection distance of 20 m or more; further, non-observation occurred for detection distances of 30 m or more. Table 4 and Table 5 show that non-observation begins to occur under intense rain (precipitation of 40 mm/h) and thick fog (visibility distance of less than 50 m) from a detection distance of 20–30 m. Compared with AL, more cases of non-observation are seen (gray shades in Table 4 and Table 5).

For PL, the NPC with a detection distance of 10 m was approximately 20 under clear conditions, but it increased under rain and fog conditions. The ratio field in Table 4 shows that this phenomenon occurred in light rain (precipitation under 20 mm/h) and fog visibility distance over 50 m within a detection distance of 30 m. As described above for point cloud plots, it seems to be caused by the fact that PL is transparent and most laser pulses pass through, whereas water droplets on the surface temporarily increase the reflectivity. However, the NPC decreased again under heavy rain (precipitation over 20 mm/h) or thick fog (visibility distance under 50 m), revealing that the increase in NPC occurs only when the weather is moderate. In other words, the weather is not strong enough for the laser pulse to scatter or be absorbed in air but is sufficient to wet the surface.

BS showed the lowest NPC value compared with that of other materials, even at the closest detection distance of 10 m, and it resulted in a significant performance drop with bad weather. This is attributable to the low reflectivity of the black surface. In Table 4 and Table 5, unlike other materials, a performance reduction could be observed even for the weakest rainfall (precipitation of 10 mm/h) and fog (visibility distance under 150 m), and unobserved phenomena continued to appear from a detection distance of 10 m.

The influence of weather conditions on the detection performance indicators differed depending on the object material and detection distance. First, within a 10 m detection distance, all materials except PL and BS were unaffected by bad weather. At 20 m, NPC reduced slightly as the fog thickened. At 30 m, NPC decreased noticeably for materials with low reflectivity, and non-observation occurred in ST with 40 mm/h precipitation and a visibility distance less than 50 m. Under these severe weather conditions, NPC of even AL decreased significantly. At 40 m or more, the reduction caused by weather conditions increased, and non-observation occurred in most materials except RF.

Considering the NPC reduction ratio shown in Table 4, the influence of the weather condition on sign detection was found to increase in the order of light rain (precipitation of 10–20 mm/h), weak fog (visibility over 50 m), intense rain (precipitation of 30–40 mm/h), and thick fog (visibility less than 50 m). However, this sequence was not always the same because somewhat different patterns were observed depending on the material.

## 6. Statistical Analysis of Performance Indicators

This section presents the changes in the performance indicators according to various weather conditions, which were tested to identify whether they were statistically significant. Accordingly, ANOVA analysis was performed to test whether any weather condition showed different performance indicators compared with those under clear conditions. Subsequently, post hoc analysis was conducted to identify the weather condition in which the performance indicator decreased.

### 6.1. NPC Differences with Different Precipitations and Equal Distance

Table 6 shows the ANOVA-analysis results for NPC and intensity by detection distance and material. The gray shadings indicate experimental combinations in which non-observation occurred (df < 7, df(res) < 32) or cases in which the F-value was significant at the 0.95 level (i.e., a group or groups in which a performance indicator decreased).

First, the NPC results (left-hand side of Table 6) indicated no statistically significant difference between clear conditions and most distance and weather conditions for RF, as in the point cloud plot and NPC results above. AL and ST started showing the effect of bad weather from a detection distance of 20 m. BS and PL, both of which have relatively low reflectivity, showed the effect of weather conditions almost all detection distances.

The results for intensity (right-hand side of Table 6) revealed that all materials were influenced by the weather.

However, for BS and PL, some combinations resulted in performance-indicator changes in the opposite direction. Examples of such combinations include NPC of BS with a detection distance of 30 m or NPC of PL with detection distances of 10 m and 40 m. Because ANOVA analysis does not reveal the weather condition for which the performance indicator changed and the direction of this change, a post hoc analysis was conducted to identify the weather condition.

### 6.2. Post Hoc Analysis of NPC and Intensity

Figure 12 and Figure 13 show the results of the post hoc experiments (Tukey’s honestly significant difference (HSD)) for NPC and intensity, respectively. This study involved many comparison groups owing to the combinations of detection distances, materials, and weather conditions. Graphs were used to present these details in a compact manner. The graphs in Figure 12 and Figure 13 present the test results for the effect of the weather conditions on each distance and material.

In each figure, several lines are drawn for each weather group; the dot symbol represents the average value in the group, and the line represents the 95% confidence interval. The clear weather condition is marked in blue, and the group in which a statistically significant difference occurs among other weather conditions is marked in red. In each figure, cases exist for which all weather conditions are not indicated. LiDAR cannot detect any points in the missing weather condition. Some graphs, such as that of BS, have lines with red shading, indicating that non-observation has already occurred (N < 5 in Table 4). These cases were classified as ones in which a performance decrease occurred (marked as “declined” in Table 6).

First, Figure 12 shows the test results for each material for NPC. In the case of RF, post hoc tests showed that there was no weather environment with statistical significance as in the ANOVA results. Even in the case of AL and ST, there was no statistically significant difference at a short detection distance (10 m). However, as the detection distance increased, the performance decreased with rain and fog.

In the case of AL, the performance decrease due to fog was significant from a detection distance of 20 m. However, the statistically significant thickness of fog was different for each detection distance. At a detection distance of 20 m, NPC reduction was significant under all fog conditions. However, from 30 m, the reduction was significant only for the thickest fog (visibility distance under 50 m). In contrast, under rainy conditions, only 40 mm/h precipitation produced a significant performance decrease at a detection distance of 30 m or more. This result is almost identical to the missing occurrence test scenario shown in Table 4, except for the fog conditions at a detection distance of 20 m.

ST showed results similar to AL. In other words, the most substantial rain (precipitation of 40 mm/h) and thickest fog (visibility less than 50 m) affected the detection distances from 20 m to 50 m. However, unlike AL, the weather conditions affecting ST included 30 mm/h and even the weakest fog (visibility distance less than 150 m), revealing its relatively low reflectivity.

In the case of BS, most weather conditions showed a statistically significant performance decrease. However, from a detection distance of 40 m, as the clear NPC itself appeared at a very low value, the number of weather conditions that showed a statistically significant difference decreased. In particular, at a detection distance of 30 m, NPC increased in response to weak rainfall (precipitation of 10 mm/h).

PL showed the most unique results at a short detection distance (10 m). As seen in the point cloud plot (Figure 10) and NPC graph above (Figure 11), a phenomenon occurred in which NPC increased during rainfall at a detection distance of 10 m. This was also revealed in the post hoc test, and the increase in NPC was significant for all precipitation and fog visibility distances of 50 m to 100 m. However, the phenomenon of NPC increase under bad weather gradually vanished from a distance of 20 m, and the performance decreased again. Most fog conditions showed a significant performance decrease from a detection distance of 20 m, and heavy rainfall (precipitation of 30–40 mm/h) also showed a significant performance decrease.

Figure 13 shows the post hoc test (Tukey’s HSD) results for intensity. In the above ANOVA test results, the f-value was significant for all experimental combinations, and the intensity decrease was expected to be significant in at least one weather condition for each combination.

First, the point cloud plot (Figure 6) and graph (Figure 11) for RF showed an insignificant intensity decrease under most weather conditions. However, an intermittent significant decrease was seen under fog conditions; therefore, the f-value was statistically significant in ANOVA results. Nevertheless, even the reduced intensity had a value of over 200, which was distinguishable from that of other materials.

Other materials such as AL, ST, BS, and PL showed decreased intensity values under most detection distances and weather conditions. Among them, AL showed more robust results than those of other materials in light rain (precipitation of 10 mm/h or 20 mm/h) and light fog (visibility over 100 m). Further, ST showed reduced performance in rainfall of over 10 mm/h and all fog conditions.

BS showed a decrease or non-observation under almost all weather conditions from a detection distance of 20 m. At the shortest detection distance of 10 m, the average intensity under clear conditions was not as high as 3.9; therefore, the decrease due to weather was not statistically significant, and only a performance decrease due to non-observation was verified.

As in the previous analysis results, the PL intensity does not change under most weather conditions at short detection distances (10 m). This tendency disappeared as the detection distance increased, and the reduction in intensity was found to be significant in the order of severe weather conditions (specifically, in order of increasing precipitation or decreasing visibility distance).

In summary, NPC and intensity decreased as the weather deteriorated, and the effect of the weather became stronger as the detection distance increased (decrease in value of indicators or non-observation under bad weather). In the case of intensity, a statistically significant decrease was confirmed for almost all materials, detection distances, and weather conditions. By contrast, in the case of NPC, despite the decrease in intensity, some cases showed no statistically significant decrease. In other words, the intensity generally decreases with deteriorating weather; however, if the reduction is insufficient, it does not lead to a reduction in NPC.

In addition, the decrease in the LiDAR-detection performance did not always occur in the order of increasing precipitation or decreasing visibility. This seems to be a limitation of the empirical experiment under actual road conditions, in which it is difficult to reproduce uniform raindrops or fog particles in the atmosphere. The result is consistent with the case in which the LiDAR-detection performance was tested in a climate chamber [20]. In the above study, the LiDAR-detection performance was significantly reduced with a smaller amount of precipitation owing to the diameter changes in water particles.

## 7. Discussion and Conclusions

In this study, an empirical test was conducted on an actual road to observe the performance degradation of LiDAR due to deteriorating weather conditions. The aim of the experiment was to identify the characteristics of LiDAR on an actual road. The results of the study are expected to contribute to the development of LiDAR-detection performance. It was observed that the detection performance of LiDAR during rain and fog conditions confirmed the claim that ‘detection performance of LiDAR degrades with deterioration of weather’, which was reported in several studies [12,13,18,19,20,21,22]. The performance of LiDAR observed in this study was indicated by the ability to detect an object, which is a key function of LiDAR. For rain conditions, we increased the precipitation from 10 to 40 mm/h in increments of 10 mm/h; for fog conditions, we decreased the visibility distance from 150 to 50 m in decrements of 50 m. The results were compared with those in clear (sunny) conditions. This study is different from the existing studies in that, in this study, rain and fog were simultaneously implemented in the same place; therefore, we could compare the results of each precipitation amount group with each visibility distance group.

The detection performance of LiDAR degraded under rain and fog conditions was compared with clear days. The performance degraded with the increase in the detection distance and decrease in the inherent reflectivity of each material. During the repeated testing, unexpected results were occasionally observed (such as NPC increase in light rain), but this was determined to be an exception.

Under rain conditions, the LiDAR characteristics observed on actual roads were identified as follows. For the condition of relatively light rain with 10–20 mm/h of precipitation, the decrease in NPC and intensity values was not significant. Therefore, LiDAR performance was adequately maintained. Moreover, considering that precipitation of 20 mm or less is the most frequent in Korea’s climate, LiDAR is suitable for use as a sensor for AVs in Korea. However, under intense rain with precipitation of 30 mm/h or more, the value of NPC decreased by a maximum of 56%, and the intensity also decreased by a maximum of 73% for RF, AL, and ST. In addition, in frequent cases, points were not collected, which indicates that LiDAR performance severely degraded. In particular, when the detection distance was more than 30 m, the performance was severely degraded for ST or other low reflective materials. When the detection distance was approximately 20 m or less under the condition of intense rain with precipitation of 30 mm or less, the decrease in the NPC and intensity value was smaller than when the detection distance was 30 m or more. This indicates that the performance of LiDAR severely degrades when detecting long-range objects under intense rain conditions. Under rain conditions, performance of LiDAR degraded less for RF objects. The result of the test case of the RF road sign indicated less performance degradation than for the sign made of other materials under intense rain of more than 30 mm/h. This result indicates that the performance of LiDAR is affected by the inherent reflectivity of the object to be detected.

Under fog conditions, the LiDAR characteristics observed on actual roads were identified as follows. In the condition of relatively weak fog, with a visibility distance between 100 and 150 m, the decrease in NPC and intensity values was not significant. However, this decrease was greater than that in the light rain conditions described above. This implies that LiDAR performance degraded more than the light rain case. However, under thick fog conditions with a visibility distance of 50 m or less, the value of NPC decreased by a maximum of 59%, and the intensity also decreased by a maximum of 71%. In addition, for this case, there were more frequent cases where points were not collected than that of the intense rain case. The amount of degradation was severest compared with the other test cases. In particular, similar to the rain condition, the degradation of performance becomes more severe when the detection distance is more than 30 m or the material is less reflective. When the detection distance was 20 m under thick fog conditions with a visibility distance of 50 m or shorter, the decrease in NPC and intensity values was minimized. The longer the detection distance, the more LiDAR performance is affected by the fog. This result is similar to the intense rain condition. However, the decrease in the performance-indicator value under the thick fog condition was more severe than that in the intense rain condition. As was the result for the rain condition test case, the performance degradation in fog was minimized for RF objects.

Because the experiment was conducted on an actual road, the results were affected not only by the artificially controlled variables (precipitation, visibility distance, detection distance, object material, etc.) but also by unpredictable variables. Therefore, the performance degradation of LiDAR was generally more severe than that in in the chamber [20]. In this study, the difference between the LiDAR’s manufacturer’s performance specifications and the characteristics of LiDAR on an actual road (particularly the performance-degradation pattern) were observed. However, there is no explanation for the degradation performance values provided by the manufacturer. These results are expected to improve the understanding of LiDAR’s use on actual roads and advance the development of AVs.

An additional motivation for conducting this study is that, in Korea, research on road signs for LiDAR detection is currently being conducted. In these studies, to improve the detection performance of LiDAR in bad weather conditions, road signs’ size, material, and arrangement are adjusted and tested. The results and the implications of this study that impact these studies are summarized as follows. To improve the road-sign visibility of LiDAR, distinguishing short- or long-distance road signs based on their purpose and selecting the size of the road signs accordingly are necessary. Furthermore, the material of the sign should have a high inherent reflectivity to maintain the performance of LiDAR. Therefore, using RF and aluminum in road sign production is appropriate. When installing a sign supporting LiDAR, the meteorological characteristics of the installation area should be considered. Because LiDAR performance deteriorates in ascending order in light rain, light fog, heavy rain, and thick fog, this pattern can be reflected in the location selection of the sign; that is, in areas where thick fog often occurs, it may be feasible to consider resetting the size of the sign and reinstalling it to maintain the performance of the LiDAR.

The results of the LiDAR-performance empirical experiments have the following limitations. This study observed the changes in NPC as a performance indicator; however, the use of these indicators has some limitations. NPC represents the number of point clouds that were acquired from the objects; however, it is insufficient to verify how adequately the original geometry is described. The geometry shape, such as that of the PL material, could not be visually identified despite high values of NPC because point clouds are concentrated only in some areas (Figure 9). Therefore, to accurately verify the LiDAR performance, it is necessary to formulate more diverse performance indicators.

In addition, to produce road signs for improving the safe driving of AVs, which is the motive of this study, the following research will be conducted in the future. For RF road signs, a blooming effect occurs (Figure 14). In this phenomenon, a point cloud larger than the actual object is formed when detecting a high reflector by LiDAR in direct sunlight [37]. Figure 13 shows the blooming effect that occurs when measuring an RF object with LiDAR. Scattered points generally have a lower Intensity value (3–10) than that of points reflected from the object surface. Scattered points should be carefully treated because they could affect the detection process. Therefore, in the future, we plan to expand the experiments to determine the cause of this blooming phenomenon and eliminate it. Lastly, to improve the visibility of road signs, not only physical improvements, such as manufacturing new signs, but also the recognition algorithms (for convergence between other sensors or different LiDAR sensors) should be studied.

## Figures and Tables

**Figure 1 sensors-23-02972-f001:**
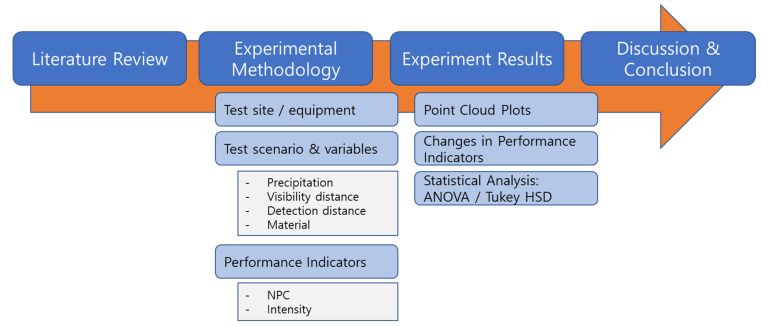
Structure of the paper.

**Figure 2 sensors-23-02972-f002:**
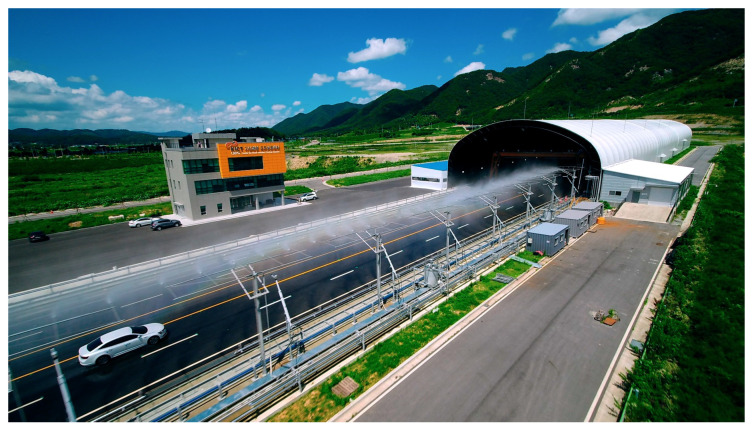
Test site at KICT Center for Road Weather Proving Ground [22].

**Figure 3 sensors-23-02972-f003:**
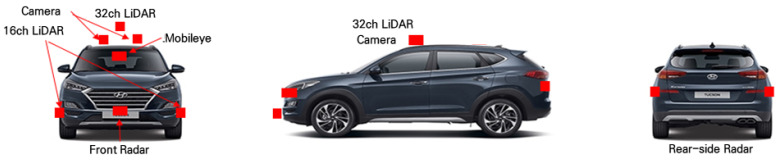
Configuration for data collection on test AV [22].

**Figure 4 sensors-23-02972-f004:**
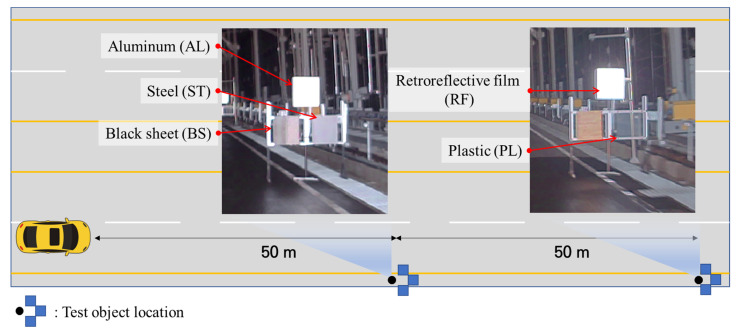
Pictures of target objects (captured from the video record file) and deployment.

**Figure 5 sensors-23-02972-f005:**
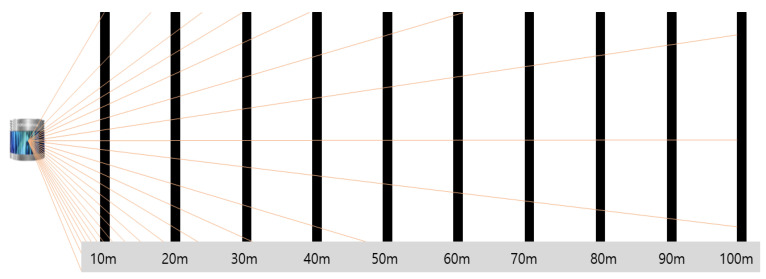
Side view of laser pulse irradiation at different distances [22].

**Figure 6 sensors-23-02972-f006:**
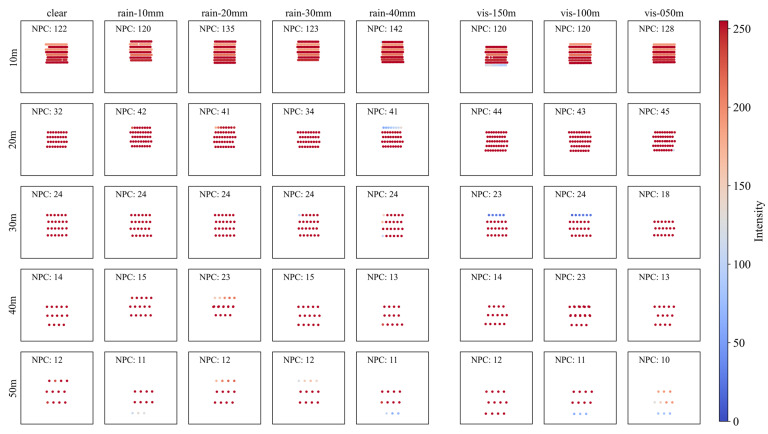
Point cloud plot of retroreflective film according to (**left**) precipitation and (**right**) visibility distance. The color of dots indicates the intensity of each point, and NPC indicates the number of points in each graph.

**Figure 7 sensors-23-02972-f007:**
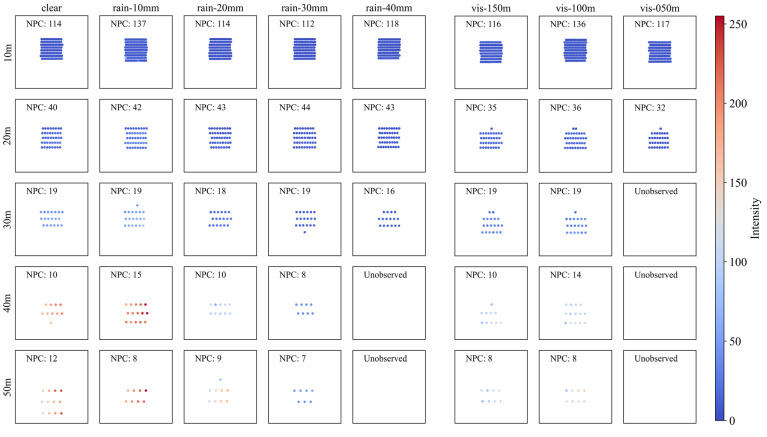
Point cloud plot of aluminum according to (**left**) precipitation and (**right**) visibility distance. The color of dots indicates the intensity of each point, and NPC indicates the number of points in each graph.

**Figure 8 sensors-23-02972-f008:**
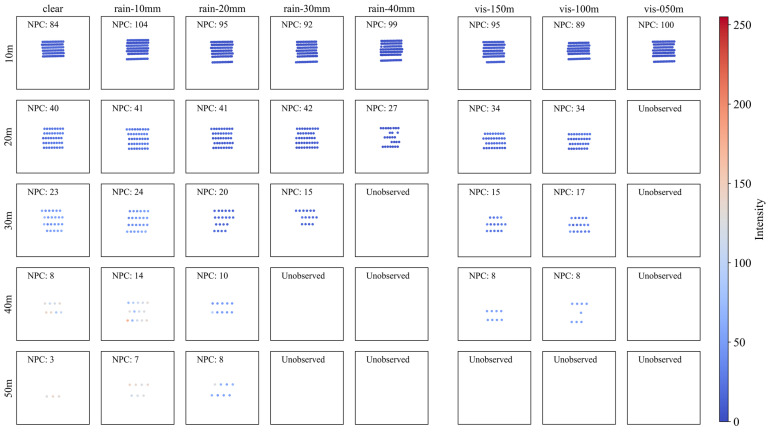
Point cloud plot of steel according to (**left**) precipitation and (**right**) visibility distance. The color of dots indicates the intensity of each point, and NPC indicates the number of points in each graph.

**Figure 9 sensors-23-02972-f009:**
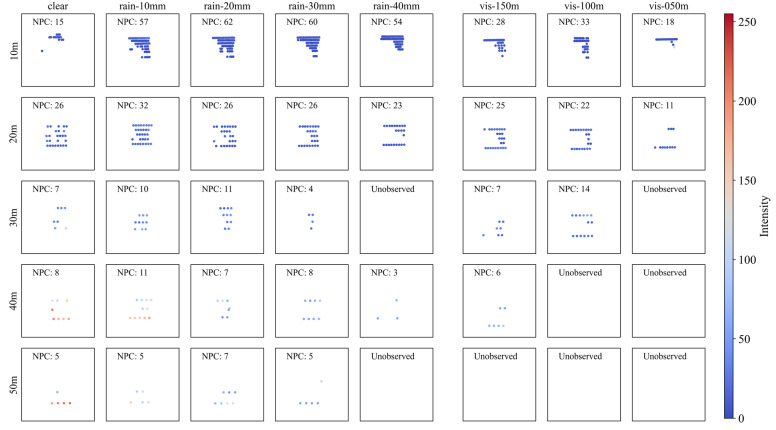
Point cloud plot of plastic according to (**left**) precipitation and (**right**) visibility distance. The color of dots indicates the intensity of each point and NPC indicates the number of points in each graph.

**Figure 10 sensors-23-02972-f010:**
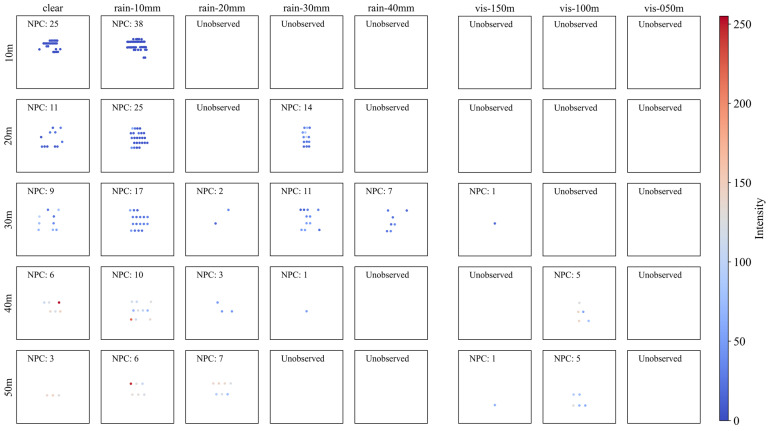
Point cloud plot of black sheet according to (**left**) precipitation and (**right**) visibility distance. The color of dots indicates the intensity of each point and NPC indicates the number of points in each graph.

**Figure 11 sensors-23-02972-f011:**
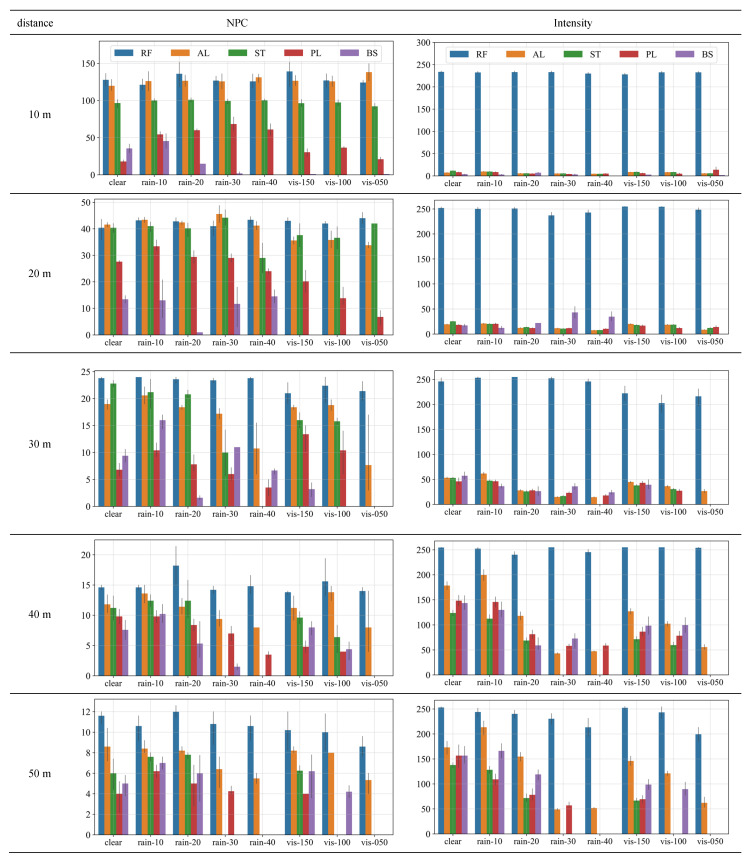
NPC and intensity according to material and weather conditions (*x*-axis). Materials: RF—Retroreflective film, AL—Aluminum, ST—Steel, BS—Black sheet, PL—Plastic. Weather conditions: rain—precipitation, vis—fog visibility distance.

**Figure 12 sensors-23-02972-f012:**
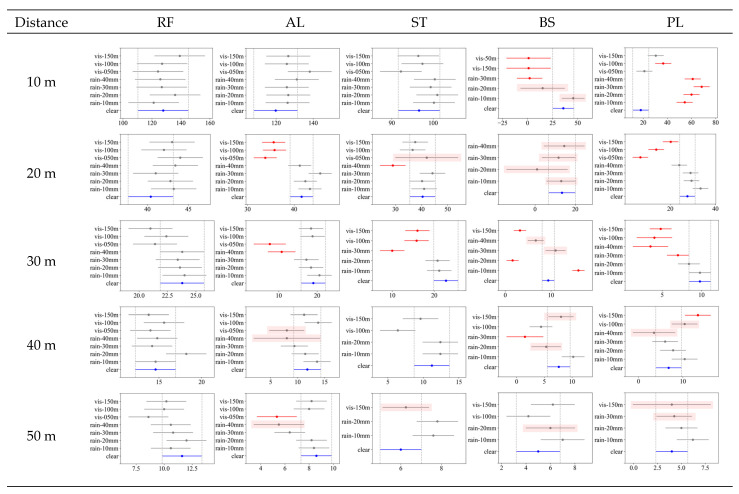
Post hoc (Tukey’s HSD) results for NPC by each material and detection distance. In each graph, the *x*-axis represents NPC, and the *y*-axis represents climate conditions. The blue lines indicate clear climate conditions. Red lines indicate climate conditions for which NPC differences are statistically significant. Gray lines indicate no difference from the NPC under clear conditions. Some graphs, such as those for BS, do not show all climate conditions or have lines with red shading, indicating that a decrease in performance indicators or non-observation has already occurred.

**Figure 13 sensors-23-02972-f013:**
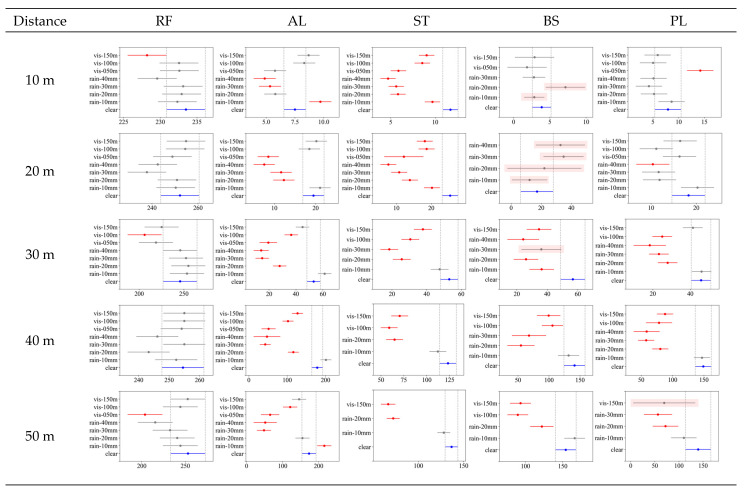
Post hoc (Tukey’s HSD) results for intensity by each material and detection distance. In each graph, the *x*-axis represents intensity, and *y*-axis represents climate conditions. Blue lines indicate clear climate conditions. Red lines indicate climate conditions for which intensity differences are statistically significant. Gray lines indicate no difference from the intensity under clear conditions. Some graphs, such as those for BS, do not show all climate conditions or have lines with red shading, indicating that a decrease in performance indicators or non-observation has already occurred.

**Figure 14 sensors-23-02972-f014:**
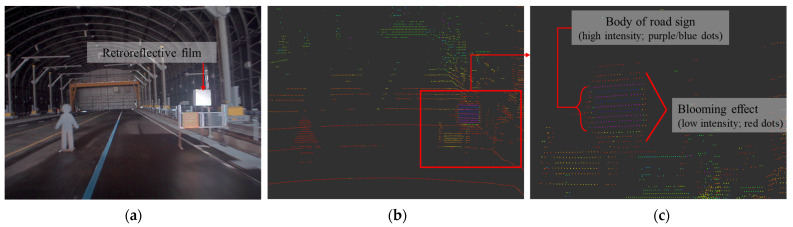
Blooming effect on retroreflective film: (**a**) test environment, (**b**) PCD from RS-32 at a detection distance of 10 m, and (**c**) enlarged view of (**b**) in which blooming effects are clearly visible.

**Table 1 sensors-23-02972-t001:** Comparison of existing studies on LiDAR’s detection performance.

Studies	Data Analysis Environment	Weather Conditions	Findings	Limitations
Kutila et al. [19]	Real road	Fog/Snow	Detection performance (detection distance) of LiDAR was reduced by 25% in fog and snowfall conditions	The amount of snowfall or fog visibility was not accurate
Goodin et al. [23]	Simulation	Rain	At the maximum rainfall of 45 mm/h, the maximum recognition distance decreased by approximately 30% (5 m) compared to a clear day, and the NPC decreased by 45%.	Results of simulations do not reflect some variables in real situations
Tang et al. [12]	Real road (parking lot)	Rain	Estimating probability of pedestrian detection significantly decreased on rainy days	The relationship between the actual amount of rainfall and the detection performance of LiDAR could not be proven
Kim et al. [22]	Real road	Rain	Distance and rainfall affect NPC and intensity Materials affect intensity. Speed does not affect the detection performance of LiDAR.	Correlation between all factors could not be confirmed

**Table 2 sensors-23-02972-t002:** Comparison of LiDAR sensors.

Performance	Robosense RS-32	Velodyne Ultra Puck	Hesai XT-32	Ouster OS1-32	Velodyne Velarray H800
Principle	Rotation	Rotation	Rotation	Rotation	Solid State
No. of channel	32	32	32	32	-
Laser wavelength	905 nm	903 nm	905 nm	865 nm	905 nm
Measurement Range	200 m	200 m	120 m	120 m	200 m
Horizontal angular resolution	0.1°–0.4° (5–20 Hz)	0.1°–0.4° (5–20 Hz)	0.18° (10 Hz)	0.18°–0.7° (10 or 20 Hz)	0.26°
Vertical angular Resolution	0.33° (minimum)	0.33°(minimum)	1°	0.35°–2.8°	0.2°–0.5°
Vertical FOV	40° (−25° to +15°)	40° (−25° to +15°)	31° (−16° to 15°)	45° (−22.5° to 22.5°)	16°(Vertical) 120°(Horizontal)
Points per second	600,000 pts/s @single return	600,000 pts/s @single return	640,000 pts/s @single return	655,360 pts/s @single return	400,000 pts/s [35] @single return

**Table 3 sensors-23-02972-t003:** Test conditions.

Factors	Control Conditions
material	white retroreflective film, aluminum, steel, plastic, and black sheet
weather	Clear rain: 10, 20, 30, and 40 mm/h fog intensity (i.e., visibility distance): 50, 100, and 150 m
detection distance	10, 20, 30, 40, and 50 m
number of drives	five for each test condition

**Table 4 sensors-23-02972-t004:** Descriptive statistics of NPC according to material and weather conditions. The ratio refers to NPC in rain or fog condition compared with that under clear conditions for each detection distance and material. Gray shading indicates NPC ratio below 1.0 (light gray) and non-observation (dark gray). Materials: RF—Retroreflective film, AL—Aluminum, ST—Steel, PL—Plastic, BS—Black sheet. Weather conditions: rain—precipitation, vis—fog visibility distance.

Detection Distance	Weather	Materials																			
		RF				AL				ST				PL				BS			
		N	Mean	Std	Ratio	N	Mean	Std	Ratio	N	Mean	Std	Ratio	N	Mean	Std	Ratio	N	Mean	Std	Ratio
10 m	clear	5	127.8	11.4		5	119.8	10.2		5	96.6	7.5		5	18.0	2.9		5	35.6	7.7	
	rain10	5	121.2	9.3	0.95	5	126.2	16.9	1.05	5	100.2	3.2	1.04	5	54.4	5.2	3.02	4	45.5	11.9	1.28
	rain20	5	136.0	31.1	1.06	5	126.6	9.7	1.06	5	101.0	3.8	1.05	5	60.0	2.5	3.33	1	15.0	-	0.42
	rain30	5	126.8	8.5	0.99	5	125.8	13.0	1.05	5	99.4	4.4	1.03	5	68.4	11.5	3.80	3	2.0	1.7	0.06
	rain40	5	125.8	11.9	0.98	5	131.2	7.5	1.10	5	100.4	2.1	1.04	5	61.0	9.7	3.39	-	-	-	-
	vis150	5	139.2	27.7	1.09	5	126.6	8.2	1.06	5	96.4	6.3	1.00	5	30.4	6.9	1.69	1	1.0	-	0.03
	vis100	5	127.0	11.9	0.99	5	125.8	9.4	1.05	5	97.4	5.1	1.01	5	36.6	2.3	2.03	-	-	-	-
	vis50	5	124.2	4.0	0.97	5	138.2	14.2	1.15	5	92.2	4.9	0.95	5	21.0	3.7	1.17	1	1.0	-	0.03
20 m	clear	5	40.4	4.9		5	41.6	1.1		5	40.4	2.2		5	27.6	0.9		5	13.4	1.8	
	rain10	5	43.2	1.1	1.07	5	43.4	1.3	1.04	5	41.0	2.4	1.01	5	33.4	2.7	1.21	4	13.0	8.8	0.97
	rain20	5	42.8	1.8	1.06	5	42.4	0.9	1.02	5	40.2	3.6	1.00	5	29.4	3.2	1.07	1	1.0	-	0.07
	rain30	5	41.0	3.9	1.01	5	45.6	4.2	1.10	5	44.2	3.7	1.09	5	29.0	2.1	1.05	3	11.7	7.8	0.87
	rain40	5	43.4	1.7	1.07	5	41.2	2.0	0.99	5	29.0	7.3	0.72	5	24.0	1.2	0.87	2	14.5	3.5	1.08
	vis150	5	43.0	1.6	1.06	5	35.6	1.9	0.86	5	37.6	5.9	0.93	5	20.2	5.7	0.73	-	-	-	-
	vis100	5	42.0	1.0	1.04	5	35.8	3.9	0.86	5	36.6	5.5	0.91	5	13.8	4.6	0.50	-	-	-	-
	vis50	5	44.0	2.7	1.09	5	33.8	1.5	0.81	1	42.0	-	1.04	5	6.8	3.4	0.25	-	-	-	-
30 m	clear	5	23.8	0.4		5	19.0	1.2		5	22.8	0.8		5	6.8	1.5		5	9.4	1.5	
	rain10	5	24.0	0.0	1.01	5	20.6	2.1	1.08	5	21.2	3.7	0.93	5	10.4	1.7	1.53	5	16.0	1.7	1.70
	rain20	5	23.6	0.9	0.99	5	18.4	0.5	0.97	5	20.8	0.8	0.91	5	7.8	1.9	1.15	5	1.6	0.5	0.17
	rain30	5	23.4	0.5	0.98	5	17.2	1.3	0.91	5	10.0	5.8	0.44	5	6.0	1.6	0.88	2	11.0	0.0	1.17
	rain40	5	23.8	0.4	1.00	4	10.8	6.0	0.57	-	-	-	-	2	3.5	2.1	0.51	3	6.7	0.6	0.71
	vis150	5	21.0	2.7	0.88	5	18.4	0.5	0.97	5	16.0	1.9	0.70	5	13.4	3.6	1.97	5	3.2	1.6	0.34
	vis100	5	22.4	3.6	0.94	5	18.8	1.3	0.99	5	15.8	0.8	0.69	5	10.4	5.1	1.53	-	-	-	-
	vis50	5	21.4	2.6	0.90	3	7.7	8.1	0.41	-	-	-	-	-	-	-	-	-	-	-	-
40 m	clear	5	14.6	0.5		5	11.8	2.0		5	11.2	2.8		5	9.8	1.5		5	7.6	2.1	
	rain10	5	14.6	0.5	1.00	5	13.6	1.9	1.15	5	12.4	1.3	1.11	5	9.8	1.3	1.00	5	10.2	1.9	1.34
	rain20	5	18.2	4.4	1.25	5	11.4	1.7	0.97	5	12.4	3.9	1.11	5	8.4	1.3	0.86	3	5.3	3.2	0.70
	rain30	5	14.2	0.8	0.97	5	9.4	1.7	0.80	-	-	-	-	5	7.0	1.6	0.71	2	1.5	0.7	0.20
	rain40	5	14.8	1.9	1.01	1	8.0	-	0.68	-	-	-	-	2	3.5	0.7	0.36	-	-	-	-
	vis150	5	13.8	0.4	0.95	5	11.2	2.7	0.95	5	9.6	1.3	0.86	5	4.8	1.3	0.49	4	8.0	1.4	1.05
	vis100	5	15.6	4.2	1.07	5	13.8	1.6	1.17	5	6.4	2.9	0.57	2	4.0	0.0	0.41	5	4.4	1.9	0.58
	vis50	5	14.0	0.7	0.96	3	8.0	5.3	0.68	-	-	-	-	-	-	-	-	-	-	-	-
50 m	clear	5	11.6	0.9		5	8.6	2.1		5	6.0	2.0		5	4.0	1.9		5	5.0	1.2	
	rain10	5	10.6	1.5	0.91	5	8.4	0.9	0.98	5	7.6	0.5	1.27	5	6.2	0.8	1.55	5	7.0	0.7	1.40
	rain20	5	12.0	0.7	1.03	5	8.2	0.4	0.95	5	7.8	0.4	1.30	5	5.0	2.5	1.25	4	6.0	2.7	1.20
	rain30	5	10.8	2.2	0.93	5	6.4	1.9	0.74	-	-	-	-	4	4.2	0.5	1.05	-	-	-	-
	rain40	5	10.6	1.5	0.91	2	5.5	0.7	0.64	-	-	-	-	-	-	-	-	-	-	-	-
	vis150	5	10.2	2.5	0.88	5	8.2	0.4	0.95	4	6.2	0.5	1.03	1	4.0	-	1.00	5	6.2	2.9	1.24
	vis100	5	10.0	2.3	0.86	5	8.0	0.0	0.93	-	-	-	-	-	-	-	-	5	4.2	0.8	0.84
	vis50	5	8.6	1.3	0.74	3	5.3	1.2	0.62	-	-	-	-	-	-	-	-	-	-	-	-

**Table 5 sensors-23-02972-t005:** Descriptive statistics of intensity according to material and weather conditions. The ratio refers to the intensity in rain or fog condition compared with that under clear conditions in each detection distance and material. Gray shading indicates intensity ratio below 1.0 (light gray) and non-observation (dark gray). Materials: RF—Retroreflective film, AL—Aluminum, ST—Steel, PL—Plastic, BS—Black sheet. Weather conditions: rain—precipitation, vis—fog visibility distance.

Detection Distance	Weather	Materials																			
		RF				AL				ST				PL				BS			
		N	Mean	Std	Ratio	N	Mean	Std	Ratio	N	Mean	Std	Ratio	N	Mean	Std	Ratio	N	Mean	Std	Ratio
10 m	clear	5	233.5	2.6		5	7.5	0.3		5	11.7	0.3		5	7.9	2.9		5	3.9	0.8	
	rain10	5	232.3	1.4	0.99	5	9.7	2.3	1.29	5	9.7	2.1	0.83	5	8.6	1.8	1.09	4	2.9	1.5	0.74
	rain20	5	232.9	2.0	1.00	5	5.8	0.8	0.77	5	5.8	0.4	0.50	5	5.2	0.8	0.66	1	7.2	-	1.85
	rain30	5	233.1	1.4	1.00	5	5.3	0.2	0.71	5	5.6	0.3	0.48	5	4.2	0.5	0.53	3	2.8	0.7	0.72
	rain40	5	229.6	4.3	0.98	5	4.9	0.4	0.65	5	4.7	0.6	0.40	5	5.1	0.4	0.65	-	-	-	-
	vis150	5	228.2	4.3	0.98	5	8.7	0.2	1.16	5	9.0	0.1	0.77	5	5.9	1.7	0.75	1	2.9	-	0.74
	vis100	5	232.6	1.1	1.00	5	8.3	0.8	1.11	5	8.5	0.9	0.73	5	5.0	0.9	0.63	-	-	-	-
	vis50	5	232.6	0.8	1.00	5	5.8	0.5	0.77	5	5.9	0.3	0.50	5	14.2	5.8	1.80	1	1.8	-	0.46
20 m	clear	5	252.2	3.8		5	19.5	0.9		5	25.4	0.7		5	18.3	1.2		5	17.2	1.3	
	rain10	5	250.3	4.7	0.99	5	21.1	5.8	1.08	5	20.2	5.2	0.80	5	20.3	4.4	1.11	4	12.2	0.6	0.71
	rain20	5	250.9	3.7	0.99	5	12.3	1.1	0.63	5	13.9	1.9	0.55	5	11.9	1.6	0.65	1	22.1	-	1.28
	rain30	5	237.8	14.5	0.94	5	11.6	0.7	0.59	5	10.8	1.0	0.43	5	11.7	1.1	0.64	3	34.9	19.0	2.03
	rain40	5	242.5	12.3	0.96	5	7.6	0.3	0.39	5	7.7	0.9	0.30	5	10.4	1.3	0.57	2	32.8	16.3	1.91
	vis150	5	254.9	0.3	1.01	5	20.2	1.7	1.04	5	18.1	0.2	0.71	5	16.4	1.7	0.90	-	-	-	-
	vis100	5	254.5	0.8	1.01	5	18.5	1.7	0.95	5	18.6	0.9	0.73	5	11.2	3.2	0.61	-	-	-	-
	vis50	5	248.9	11.0	0.99	5	8.6	2.8	0.44	1	12.1	-	0.48	5	16.3	7.9	0.89	-	-	-	-
30 m	clear	5	245.8	20.2		5	53.2	2.3		5	53.0	1.1		5	45.7	5.1		5	56.7	4.8	
	rain10	5	253.5	2.7	1.03	5	61.3	5.8	1.15	5	47.5	0.8	0.90	5	45.9	3.2	1.00	5	36.3	5.5	0.64
	rain20	5	255.0	0.0	1.04	5	27.9	6.1	0.52	5	25.6	9.1	0.48	5	27.9	4.3	0.61	5	26.1	8.0	0.46
	rain30	5	252.4	2.7	1.03	5	14.9	2.0	0.28	5	18.4	6.8	0.35	5	23.3	1.5	0.51	2	36.1	0.8	0.64
	rain40	5	246.0	12.4	1.00	4	14.1	1.2	0.27	-	-	-	-	2	18.4	1.8	0.40	3	24.2	3.3	0.43
	vis150	5	225.1	27.5	0.92	5	44.9	5.9	0.84	5	37.8	5.7	0.71	5	41.3	8.6	0.90	5	34.6	13.5	0.61
	vis100	5	206.1	27.7	0.84	5	36.5	1.5	0.69	5	30.6	1.0	0.58	5	25.0	5.9	0.55	-	-	-	-
	vis50	5	218.8	25.5	0.89	3	19.4	10.1	0.36	-	-	-	-	-	-	-	-	-	-	-	-
40 m	clear	5	254.5	1.0		5	178.5	5.0		5	123.2	2.5		5	149.0	11.3		5	141.3	15.3	
	rain10	5	252.3	3.7	0.99	5	201.1	20.8	1.13	5	112.2	7.9	0.91	5	147.0	22.8	0.99	5	131.6	17.9	0.93
	rain20	5	243.3	16.0	0.96	5	116.1	21.8	0.65	5	64.9	16.7	0.53	5	80.6	7.1	0.54	3	54.8	10.7	0.39
	rain30	5	255.0	0.0	1.00	5	42.7	3.3	0.24	-	-	-	-	5	58.0	3.7	0.39	2	67.8	21.3	0.48
	rain40	5	246.2	9.3	0.97	1	47.2	-	0.26	-	-	-	-	2	58.7	0.7	0.39	-	-	-	-
	vis150	5	255.0	0.0	1.00	5	127.0	14.0	0.71	5	70.5	9.5	0.57	5	88.1	11.8	0.59	4	99.4	17.0	0.70
	vis100	5	255.0	0.0	1.00	5	102.1	5.1	0.57	5	59.0	5.3	0.48	2	78.4	0.0	0.53	5	105.7	18.3	0.75
	vis50	5	254.1	2.0	1.00	3	51.4	9.7	0.29	-	-	-	-	-	-	-	-	-	-	-	-
50 m	clear	5	253.0	2.5		5	173.4	17.4		5	136.7	6.8		5	139.9	46.0		5	154.1	14.1	
	rain10	5	244.4	15.2	0.97	5	215.9	30.7	1.25	5	128.1	6.5	0.94	5	110.3	17.9	0.79	5	166.0	19.5	1.08
	rain20	5	240.7	12.4	0.95	5	155.1	15.9	0.89	5	72.3	11.8	0.53	5	72.2	15.8	0.52	4	122.0	8.8	0.79
	rain30	5	232.5	14.1	0.92	5	48.4	4.0	0.28	-	-	-	-	4	56.5	6.1	0.40	-	-	-	-
	rain40	5	215.5	20.9	0.85	2	51.7	0.2	0.30	-	-	-	-	-	-	-	-	-	-	-	-
	vis150	5	252.7	4.4	1.00	5	145.9	24.8	0.84	4	66.7	0.3	0.49	1	69.5	-	0.50	5	93.4	19.1	0.61
	vis100	5	244.3	22.7	0.97	5	121.1	5.0	0.70	-	-	-	-	-	-	-	-	5	89.7	4.4	0.58
	vis50	5	203.8	38.3	0.81	3	65.4	22.6	0.38	-	-	-	-	-	-	-	-	-	-	-	-

**Table 6 sensors-23-02972-t006:** ANOVA results for NPC and intensity. Gray shading indicates experimental combination that was statistically significant or that resulted in non-observation. Materials: RF—Retroreflective film, AL—Aluminum, ST—Steel, BS—Black sheet, PL—Plastic.

Detection Distance	Material	NPC					Intensity				
		df	df(res)	F	PR (>F)	Result	df	df(res)	F	PR (>F)	Result
10 m	RF	7	32	0.63	0.729	-	7	32	2.69	0.026	declined
	AL	7	32	1.05	0.416	-	7	32	18.29	0.000	declined
	ST	7	32	1.77	0.128	-	7	32	43.27	0.000	declined
	PL	7	32	45.83	0.000	increased	7	32	8.56	0.000	increased
	BS	5	9	13.72	0.001	declined	5	9	3.62	0.045	declined
20 m	RF	7	32	1.06	0.409	-	7	32	2.64	0.028	declined
	AL	7	32	15.71	0.000	declined	7	32	24.10	0.000	declined
	ST	7	28	4.57	0.002	declined	7	28	32.52	0.000	declined
	PL	7	32	35.89	0.000	declined	7	32	5.44	0.000	declined
	BS	4	10	0.96	0.469	declined	4	10	3.08	0.068	declined
30 m	RF	7	32	1.93	0.097	-	7	32	5.00	0.001	declined
	AL	7	29	8.01	0.000	declined	7	29	63.32	0.000	declined
	ST	5	24	12.91	0.000	declined	5	24	31.67	0.000	declined
	PL	6	25	5.10	0.002	declined	6	25	20.84	0.000	declined
	BS	5	19	75.31	0.000	declined	5	19	9.57	0.000	declined
40 m	RF	7	32	1.88	0.106	-	7	32	2.34	0.048	declined
	AL	7	26	2.98	0.020	declined	7	26	77.57	0.000	declined
	ST	4	20	4.53	0.009	declined	4	20	46.69	0.000	declined
	PL	6	22	13.78	0.000	declined	6	22	41.76	0.000	declined
	BS	5	18	7.66	0.001	declined	5	18	14.40	0.000	declined
50 m	RF	7	32	1.78	0.126	-	7	32	4.15	0.002	declined
	AL	7	27	4.21	0.003	declined	7	27	43.74	0.000	declined
	ST	3	15	3.25	0.051	declined	3	15	103.18	0.000	declined
	PL	4	15	1.29	0.317	declined	4	15	7.05	0.002	declined
	BS	4	19	1.67	0.199	declined	4	19	27.82	0.000	declined

## Data Availability

Not applicable.

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
