# Peer review of "Empirical Analysis of Autonomous Vehicle’s LiDAR Detection Performance Degradation for Actual Road Driving in Rain and Fog"

_sensors, 2023, doi:10.3390/s23062972_

Round 1

Reviewer 1 Report

This paper provides a thorough examination of the performance degradation of LIDAR detection in autonomous vehicles during real road driving under adverse weather conditions such as rain and for. The use of empirical data adds credibility to the findings, demonstrating the need for continued research and development in the areas to ensure the safety and reliability to autonomous vehicles. A well written and insightful contribution to the field.

 minor Changes are mentioned as below:

In section "5.1. Overview" author have mentioned "In this chapter". Replace it with "In this paper".

Re-write the last paragraph(Line no.665-672)

Reviewer 2 Report

Dear authors,

       you have conducted an empirical study about LiDAR sensing performance in adverse conditions. The reviewer has some suggestions and comments:

1. Many of the figures and the texts in the figures are very blurry. Please improve the quality of the figures.

2. In Table 1, would you consider also discussing solid-state LiDAR sensors? Can your study be suitable for them?

3. Some tables contain many results. Would you consider better highlighting some results?

4. How about a combination of different sensors? Please discuss this.

5. Instead of a discrete setting of the degradation levels, would you consider a more continuous setting? Please discuss this.

6. Some recent robust LiDAR segmentation works could be disucssed. [*] "Benchmarking the Robustness of LiDAR Semantic Segmentation Models." arXiv preprint arXiv:2301.00970 (2023). [*] "MASS: Multi-attentional semantic segmentation of LiDAR data for dense top-view understanding." IEEE Transactions on Intelligent Transportation Systems 23.9 (2022): 15824-15840. 

For these reasons, a revision is recommended before this paper can be considered for publication.

Sincerely,

Reviewer 3 Report

Comments for Authors

-          The English of the paper is not clear in several parts, and some parts are not clear enough to understand the authors' idea. The English should be improved and the grammatical mistakes should be corrected.

-          A flow chart of the developed method is needed to make it more understandable.

-          Please ensure consistency in using different key terms.

-          Please improve the introduction section by providing the background, gaps, and contributions of the study in a more specific way. A couple of examples from real-life examples would enhance the motivation of the study.

-          The literature review should be enhanced by presenting a critical review, not just presenting information about who did what. The authors should prepare a table to highlight the previous contributions and research gaps in a more robust way. The recent references must be cited and explained. The references from top journals should be explored.

-          Identifying key factors should not be a part of the literature review. It should be a part of the result analysis. Literature review analyses the closely related papers to identify research gaps.

-          The method section should be enhanced by presenting more clear details. It is hard to understand some parts and also, the integrity and consistency of the method is hard to understand.

-          The result analysis should be improved based on the unique findings, interesting insights, and how these results will be useful to the practice.

-          The managerial implications should be provided based on numerical results and findings. How would the manager be benefited from the findings of the study? What are the specific action plans based on the research findings?

-          The conclusion section can be revised considering unique findings, contributions, limitations, and future research directions.

-          Check the citations and references (one by one) if there is any missing information. Citations and references must be 100% accurate.

Round 2

Reviewer 3 Report

-          The English of the paper still is not clear in several parts, and some parts are not clear enough to understand the authors' idea. The English should be improved and the grammatical mistakes should be corrected.

-          The conclusion section can be revised and rewrite in more efficient.

Author Response

Dear reviewer,

Thank you for reviewing our manuscript.

As you mentioned, the manuscript had a lot of room for improvement.

The responses to your comments are as follows:

  1. The English of the paper still is not clear in several parts, and some parts are not clear enough to understand the authors' idea. The English should be improved and the grammatical mistakes should be corrected.

We have reviewed the entire manuscript and corrected grammatical errors and clarified statements with inaccurate meanings. In addition, we requested a proofreading from a native English speaker for the manuscript.

We attached a certificate of editing.

  1. The conclusion section can be revised and rewrite in more efficient.

We agree with your comment. Section 7 was rather long and complicated to read.

Reviewing the contents of Section 7 again, we removed some parts and refined the writings more efficiently. In addition, indentation and paragraph breaks have been adjusted to improve readability.

Sincerely,

Bumjin Park
